# Accurate Performance Analysis of Coded Large-Scale Multiuser MIMO Systems with MMSE Receivers

**DOI:** 10.3390/s19132884

**Published:** 2019-06-28

**Authors:** Kai Zhai, Zheng Ma, Xianfu Lei

**Affiliations:** Key Lab of Information Coding and Transmission, Southwest Jiaotong University, Chengdu 610031, China

**Keywords:** large-scale multi-user MIMO systems, SINR distribution, MMSE detector, convolutional code, performance analysis

## Abstract

In this paper, we estimate the uplink performance of large-scale multi-user multiple-input multiple-output (MIMO) networks. By applying minimum-mean-square-error (MMSE) detection, a novel statistical distribution of the signal-to-interference-plus-noise ratio (SINR) for any user is derived, for path loss, shadowing and Rayleigh fading. Suppose that the channel state information is perfectly known at the base station. Then, we derive the analytical expressions for the pairwise error probability (PEP) of the massive multiuser MMSE–MIMO systems, based on which we further obtain the upper bound of the bit error rate (BER). The analytical results are validated successfully through simulations for all cases.

## 1. Introduction

Large-scale multiuser multiple-input multiple-output (MU-MIMO) systems are characterized by consisting of a hundred, or even more antennas at the base station (BS) and multiple users being served simultaneously, which dramatically improves the link reliablility and spectral efficiency of wireless communication systems [1]. More attractively, combining with channel coding techniques and linear signal processing technologys, e.g., Zero-Forcing (ZF) and Minimum-Mean-Square-Error (MMSE) can further enhance the performance of massive MIMO. With the appropriate user scheduling scheme [2], all communicating entities can be obtained an equal opportunity without sacrificing sum-rate performance. For MU-MIMO uplink orthogonal frequency division multiple access (OFDMA) video transmission systems, the authors of [3] propose an average peak-signal-to-noise-ratio (PSNR) optimized cross-layer resource allocation and user grouping scheme, in which zero-forcing MU-MIMO detector is considered. When the number of scatterers is small compared to the base station antenna and single antenna users in the cell, the authors of [4] proposes a fast iterative weighted singular value thresholding (WSVT) algorithm to obtain a more accurate channel estimate.It is worth emphasizing that, when the elements of the BS antenna array become large, the column-vectors of the propagation matrix from BS to users tend to be orthogonal [5,6]. Considering a ZF detector, an approximate distribution for the received SINR and an approximated symbol error probability (SEP) were explored [7]. When an MMSE receiver with beamforming is considered in massive MIMO links, the asymptotic evaluation of the target SINR was investigated in [8]. Work that relates directly to the throughput bound for convolutionally encoded large-scale MIMO-OFDM links was investigated in [9], by using the approximate noise distribution for the ZF detector. Moreover, the energy and spectral efficiency of the uplink large multiuser MIMO wireless networks are analyzed in [10]. We note that the authors only discuss the rate by a lower bound result, based on the approximated gamma distribution of SINR for the MMSE receiver considered in [11]. According to the exact SINR distribution provided in [12], the authors of [13] studied the performance of convolutionally encoded MMSE–MIMO Rayleigh fading channels. However, for large-scale multi-user MIMO systems, this distribution cannot be analyzed analytically. By considering additive quantization noise model (AQNM), the authors of [14] studied the MMSE detection performance of a hybrid analog-to-digital converters (ADC) massive MIMO system, which can reduce energy consumption without reducing spectral efficiency compared to a full-resolution ADC system. Impressively, a system named cell-free massive MIMO with many distributed access points was investigated in [15]; by applying MMSE detection, the author comprehensively analyzed the spectral efficiencies of the system, and the analysis results showed that the cell-free massive MIMO is a more promising technology than conventional Cellular massive MIMO in beyond-5G networks. Furthermore, a closed-form distribution for the output SINR of the MMSE receiver was derived in [16], for uncoded MIMO systems in ideal fast Rayleigh fading, and accurate symbol error performance was evaluated. In this paper, we focus on the investigation of the uplink performance for coded cellular large-scale multiuser MIMO systems with an MMSE receiver. Our research work so far makes the following specific contributions:We derive a novel statistical distribution of the SINR for any user when the channels are subject to path loss, shadowing and Rayleigh fading.Using the derived SINR density, we further derive the PEP, which is then used to obtain analytical expressions relating to upper-bounds on BER of the convolutionally coded massive multiuser MMSE–MIMO systems.

Compared with [16], the channel considered in this paper is more in line with the actual multi-user MIMO scenario. In addition, we extend the analysis results to the encoded case.

The rest of this paper is structured as follows: Section 2 introduces the model of multi-user MIMO system with MMSE equalizer. Section 3 derives the corresponding SINR distribution expression. The derivation of the PEP and the upper-bounds of the BER for the convolutionally coded massive multiuser MMSE–MIMO systems are introduced in Section 4. Numerical simulations were performed in Section 5 to verify our analytical results. Finally, Section 6 summarizes the paper.

Notations: The bold lower and upper case letters denote column vectors and matrices, respectively, and ${\left[\cdot \right]}^{T},{\left[\cdot \right]}^{H}$ denotes transpose and conjugate transpose, respectively. The short-hand form CN(μ,Σ) denotes the probability density function (PDF) of a complex Gaussian random variable (RV) with mean μ and variance Σ; I is the identity matrix. The operation $E_{x}\left[f\left(x\right)\right]$ represents the mean of the function f(x) with respect to the RV *x*.

## 2. System Model

Considering the uplink of a multiuser massive MIMO system, the system consisting of one BS and *K* single-antenna users. The BS are equipped with *R* antennas, i.e., R>K. We assume that single antenna users are independent of each other. The *K* transmit symbols of the users can be organized into a K×1 vector $x=[x_{1},\cdots ,x_{K}]$, which is normalized such that $E[xx^{H}]=I$. For a given time instant, the received R×1 signal vector at BS reads
(1)$$\begin{array}{c} y=\sqrt{\frac{E_{s}}{K}}Gx+n, \end{array}$$
where $G=HD^{\frac{1}{2}}$ represents the R×K propagation matrix from the BS to the *K* users. The elements $h_{r,k}∼\mathrm{CN}\left(0,1\right)$, r=1,2,⋯,R and k=1,2,⋯,K, of the R×K channel matrix H denotes uncorrelated ideal fast fading coefficients from the BS to the *K* users. *K*-dimensional diagonal matrix $D^{\frac{1}{2}}=\mathrm{diag}\left\{\sqrt{\zeta_{1}},\sqrt{\zeta_{2}},\cdots ,\sqrt{\zeta_{K}}\right\}$ models the geometric attenuation and shadow fading. Furthermore, $\sqrt{\zeta_{k}}$ is independent of parameter *r* and changes slowly over time, which can be assumed to be a constant [10,17]. Moreover, the average transmit power per user is $\frac{E_{s}}{K}$, and $n∼\mathrm{CN}(0,N_{0}I)$, is the R×1 additive white Gaussian noise. Then, the average received SNR is given by
(2)$$\begin{array}{c} \mathrm{SNR}=\frac{E_{s}}{K}\frac{E[||HD^{\frac{1}{2}}{x]||}^{2}}{E[||n|{|}^{2}]}=\frac{E_{s}\mathrm{tr}\left(P\right)}{KN_{0}}. \end{array}$$

The choice of MMSE filtering matrix is given by the following well-known formula
(3)$$\begin{array}{c} W={\left(G^{H}G+\rho I\right)}^{-1}G^{H}, \end{array}$$
where $\rho =\frac{KN_{0}}{E_{s}}$.

Analogous to [18] (Equation (Equation 10)), with the MMSE detector, the SINR for the *k*-th data stream is expressed as
(4)$$\begin{array}{cc} \gamma_{k} & =\zeta_{k}h_{k}^{H}{\left(\sum_{r=1,r\neq k}^{K}\zeta_{r}h_{r}h_{r}^{H}+\rho I_{R}\right)}^{-1}h_{k}\\ & \overset{\left(a\right)}{=}\zeta_{k}{\tilde{h}}_{k}^{H}\Lambda {\tilde{h}}_{k}, \end{array}$$
where (a) of Equation (Equation 4) is obtained by the eigenvalue decomposition of the matrix $\sum_{r=1,r\neq k}^{K}\zeta_{r}h_{r}h_{r}^{H}$. $h_{k}$ is the *k*-th column of H, ${\tilde{h}}_{k}={\left[{\tilde{h}}_{1,k},\ldots ,{\tilde{h}}_{R,k}\right]}^{T}$ has the same distribution as $h_{k}$, and
(5)$$\begin{array}{c} \Lambda =\mathrm{diag}\left(\frac{1}{\lambda_{k,1}+\rho },\ldots ,\frac{1}{\lambda_{k,K-1}+\rho },\frac{1}{\rho },\ldots ,\frac{1}{\rho }\right). \end{array}$$

Moreover, $\lambda_{k,1},\cdots ,\lambda_{k,K-1}$ are the K−1 non-zero eigenvalues of the matrix $\sum_{r=1,r\neq k}^{K}\zeta_{r}h_{r}h_{r}^{H}$. Therefore, ${\mathrm{SINR}}_{i}$ can be represented as
(6)$$\begin{array}{cc} \gamma_{k} & =\zeta_{k}\sum_{i=1}^{K-1}\frac{|{\tilde{h}}_{i,k}{|}^{2}}{\lambda_{k,i}+\rho }+\zeta_{k}\sum_{j=K}^{R}\frac{|{\tilde{h}}_{j,k}{|}^{2}}{\rho }. \end{array}$$

## 3. Distribution of the SINR

As shown, in Refs. [11,18,19], the problem encountered in performance analysis of MMSE-detected MIMO systems is to explore an efficient approach to evaluate Equation (Equation 6).

In this section, we develop a simple and efficient way to derive the distribution expression for SINR in massive multiuser MIMO scenarios. To do that, firstly, we consider the expression for the SINR in Equation (Equation 6). The term $\sum_{r=1,r\neq k}^{K}\zeta_{r}h_{r}h_{r}^{H}$ is a R×R Hermitian matrix that can be reformulated as $H_{k}D_{kk}H_{k}^{H}$, where $H_{k}$ is a R×(K−1) matrix by removing the *k*-th column of H and $D_{kk}$ corresponds to D with the *k*th row and the *k*th column deleted.

We note that Equation (Equation 6) contains K−1 different nonzero eigenvalues to be solved. The matrix $H_{k}D_{kk}H_{k}^{H}$ is a complex Wishart matrix; let ${\tilde{\zeta }}_{1},\cdots ,{\tilde{\zeta }}_{K-1}$ be the ordered eigenvalues of the diagonal matrix $D_{kk}$ such that ${\tilde{\zeta }}_{1}>\cdots >{\tilde{\zeta }}_{K-1}>0$.

The joint PDF of the ordered positive eigenvalues of $H_{k}D_{kk}H_{k}^{H}$, $\lambda_{1},\cdots ,\lambda_{K-1}$, equals [20]
(7)$$\begin{array}{c} f_{\lambda }=L\det\left[E\right]\prod_{l=1}^{K-1}\lambda_{l}^{R-K+1}\prod_{k<l}^{K-1}\left(\lambda_{k}-\lambda_{l}\right), \end{array}$$
where $E={\left\{e^{-\lambda_{k}/{\tilde{\zeta }}_{l}}\right\}}_{k,l=1,\cdots ,K-1,}$
(8)$$\begin{array}{c} L=\prod_{l=1}^{K-1}\frac{1}{{\tilde{\zeta_{l}}}^{R}\left(R-l\right)!}\prod_{k<l}^{K-1}\frac{{\tilde{\zeta }}_{k}{\tilde{\zeta }}_{l}}{{\tilde{\zeta }}_{k}-{\tilde{\zeta }}_{l}}. \end{array}$$

As illustrated in [19], the joint distribution of non-zero eigenvalues is directly used without approximating, and, even if the dimension of the matrix is less than 4, it is difficult to obtain a closed solution of the SINR distribution. Influenced by the large-scale fading factor, the eigenvalue joint distribution function of $HDH^{H}$ is more complicated than the eigenvalue joint distribution function of $HH^{H}$. Therefore, method [16] is not suitable for Equation (Equation 6).

Here, we employ the approximate marginal PDF of the eigenvalues to evaluate Equation (Equation 6).

The marginal PDF of an unordered eigenvalue of $H_{k}D_{kk}H_{k}^{H}$ is [21]
(9)$$\begin{array}{c} f\left(\lambda \right)=\frac{L}{K-1}\sum_{i=1}^{K-1}\sum_{j=1}^{K-1}D_{i,j}\lambda^{R-K+j}e^{-\lambda /\tilde{\zeta_{i}}}, \end{array}$$
where $D_{i,j}$ is the (i,j)-th cofactor of the matrix
(10)$$\begin{array}{c} D={\left\{\frac{(R-K+k)!}{{\tilde{\zeta }}_{l}^{-R+K-k-1}}\right\}}_{l,k=1,\cdots ,K-1}. \end{array}$$

Then, the PDF of $\gamma_{k}$ conditioned on H can be written as follows:(11)$$\begin{array}{c} f_{\gamma_{k}}\left(x|H\right)=\zeta_{k}\underset{G}{\underset{⏝}{E_{\lambda }\left[\frac{1}{\lambda_{i,k}+\rho }\right]}}\sum_{i=1}^{K-1}|{\tilde{h}}_{i,k}{|}^{2}+\frac{\zeta_{k}}{\rho }\sum_{j=K}^{R}{\left|{\tilde{h}}_{j,k}\right|}^{2}, \end{array}$$
where G can be calculated as follows: (12)$$\begin{array}{cc} G & =E\left[\frac{1}{\lambda +\rho }\right]=\int_{0}^{\infty }\frac{1}{\lambda +\rho }f\left(\lambda \right)d\lambda \\ & =\frac{L}{K-1}\sum_{i=1}^{K-1}\sum_{j=1}^{K-1}D_{i,j}\int_{0}^{\infty }\frac{1}{\lambda +\rho }\lambda^{R-K+j}e^{-\lambda /{\tilde{\zeta }}_{i}}d\lambda \\ & =\frac{L}{K-1}\sum_{i=1}^{K-1}\sum_{j=1}^{K-1}D_{i,j}\Gamma \left(R-K+j+1\right)\int_{0}^{\infty }e^{-\rho t}{\left(t+\frac{1}{{\tilde{\zeta }}_{i}}\right)}^{-R+K-j-1}dt\\ & =\frac{L}{K-1}\sum_{i=1}^{K-1}\sum_{j=1}^{K-1}D_{i,j}\Gamma \left(R-K+j+1\right)\\ & \times \left[\frac{1}{(-R+K-j-2)!}\sum_{t=1}^{-R+K-j-2}\left(t-1\right)!{\left(-\rho \right)}^{-R+K-j-2-t}{\left(\frac{1}{{\tilde{\zeta }}_{i}}\right)}^{-t}-\frac{{\left(-\rho \right)}^{-R+K-j-2}}{(-R+K-j-2)!}e^{\frac{\rho }{{\tilde{\zeta }}_{i}}}\mathrm{Ei}\left(-\frac{\rho }{{\tilde{\zeta }}_{i}}\right)\right], \end{array}$$
where we have used [22] (Equation 3.353) for the final equality. Ei(x) represents the exponential integral $\int_{-\infty }^{x}\frac{e^{t}}{t}dt$. Since all the non-zero eigenvalues have the same edge distribution, G is equivalent to the mean of $\frac{1}{\lambda +\rho }$, where the subscripts of the eigenvalues is emitted. Therefore, we can also use the statistics of λ to obtain G.

In theory, for matrix $H_{k}D_{kk}H_{k}^{H}$ with arbitrary dimensions, we can use edge distribution (Equation 9) of the eigenvalues or statistics to obtain G. However, because the edge distribution function is relatively complicated, the parameter G becomes more and more difficult to obtain when the size of the matrix becomes large.

In order to solve this problem, we first introduce an important concept from [23], that is, for an n×n Hermitian matrix A with empirical spectral/eigenvalue distribution $F_{n}$, which η-transform is defined as
(13)$$\begin{array}{c} \eta_{n}\left(\varphi \right)=\int \frac{1}{1+\varphi x}dF_{n}\left(x\right), \end{array}$$
where φ is a nonnegative real number.

In addition, for the high-dimensional diagonal matrix $D_{kk}$, we can consider replacing the diagonal elements with the first moment of the empirical spectral distribution [20] (Equation 1.3.2) of $D_{kk}$ to reduce the computational complexity.

Obviously, $H_{k}$ and $D_{kk}$ are independent. In contrast to Theorem
2.39 in [23], when K−1 satisfies (K−1)/R→c>0 as R→∞, then, almost surely, the empirical spectral distribution of $H_{k}D_{kk}H_{k}^{H}$ converges to a distribution whose η-transform can be written as
(14)$$\begin{array}{c} \eta_{H_{k}D_{kk}H_{k}^{H}}\left(\varphi \right)=1-\frac{F(\bar{\zeta }\varphi ,c)}{4\bar{\zeta }\varphi }, \end{array}$$
where
(15)$$\begin{array}{c} F\left(u,z\right)={\left(\sqrt{u{\left(1+\sqrt{z}\right)}^{2}+1}-\sqrt{u{\left(1-\sqrt{z}\right)}^{2}+1}\right)}^{2}, \end{array}$$
and $\bar{\zeta }=\sum_{i=1,i\neq k}^{K}\zeta_{i}$.

Replacing the parameter φ in Equation (Equation 14) with 1/ρ, we can easily get
(16)$$\begin{array}{c} G=\frac{1}{\rho }\left(1-\frac{F(\bar{\zeta }/\rho ,c)}{4\bar{\zeta }/\rho }\right). \end{array}$$

Since the meaning of the parameter is the same, for the convenience of future use, here, we still let G represent the expectation values in Equation (Equation 11).

Up to now, we have derived parameters G that are suitable for traditional multiuser MIMO systems and large-scale multiuser MIMO systems, respectively.

Let $\beta ={\left[\beta_{1},\cdots ,\beta_{R}\right]}^{T}$, where $\beta_{r}$ is the coefficient of $|{\tilde{h}}_{r}{|}^{2}$ in Equation (Equation 11). Note that two of the *R* entries are distinct in β. Let’s divide the elements $\beta_{1},\cdots ,\beta_{R}$ into two groups; each group is the collection of the entries equal to the same value, respectively. Moreover, let $t={\left[t_{1},t_{2}\right]}^{T}$, whose entries are the number of elements in each group, i.e., $t_{1}=K-1,t_{2}=R-K+1$. Similar to that in [24], we define the following parameters:(17)$$\begin{array}{c} A=\prod_{j=1}^{2}\delta_{j}^{-t_{j}}={\delta_{1}}^{-t_{1}}{\delta_{2}}^{-t_{2}}, \end{array}$$
(18)$$\begin{array}{cc} B_{m,l,r,\delta }={\left(-1\right)}^{l+1} & \sum_{i\in \Omega_{m,l}}\prod_{j=1,j\neq m}^{2}\left(\begin{array}{c} i_{j}+t_{j}-1\\ i_{j} \end{array}\right){\left(\frac{1}{\delta_{j}}-\frac{1}{\delta_{m}}\right)}^{-(t_{j}+i_{j})}, \end{array}$$
where the values of $\delta_{1}$ and $\delta_{2}$ are equal to $\zeta_{k}G$ and $\zeta_{k}/\rho$, respectively. The vector $i={\left[i_{1},i_{2}\right]}^{T}$ is extracted from the set $\Omega_{m,l}$ of all nonnegative integer partitions of l−1 (with $i_{m}=0$). The set $\Omega_{m,l}$ is defined as
(19)$$\begin{array}{cc} \Omega_{m,l}=\{i={\left[i_{1},i_{2}\right]}^{T}\in Z^{2}; & \sum_{j=1}^{2}i_{j}=l-1,i_{m}=0,i_{j}\geq 0\;\forall j\}. \end{array}$$

Since ${\tilde{h}}_{k}∼\mathrm{CN}\left(0,1\right)$, $|{\tilde{h}}_{k}{|}^{2}$ is an exponential RV with unit mean. Then, the right-hand side of Equation (Equation 11) is the sum of *R* independent exponential RVs with different means, which follows Generalized chi-squared distribution [24]. Therefore, the PDF of γ is given by
(20)$$\begin{array}{cc} f_{\gamma }\left(x\right) & =A\sum_{m=1}^{2}\sum_{l=1}^{t_{m}}\frac{B_{m,l,t,\delta }}{(t_{m}-l)!}x^{t_{m}-l}e^{-\frac{x}{\delta_{m}}}. \end{array}$$

In the following section, based on the derived SINR distribution, we will discuss the performance of the coding massive multiuser MIMO system.

## 4. Performance Analysis of Coded Multiuser MIMO System with an MMSE Detector

In this section, we derive an upper bound on the BER of the encoded large-scale multiuser MMSE–MIMO systems that employ an *M*-ary quadrature amplitude modulation (QAM) constellation with gray mapping.

By using an MMSE Receiver in massive MIMO systems, at the base station, the decision variable $z_{k}$ for the transmission sequence $x_{k}$ can be expressed as
(21)$$\begin{array}{c} z_{k}=\xi_{k}x_{k}+\eta_{k},\;\;\;1\leq k\leq K, \end{array}$$
where $\xi_{k}=\sqrt{\frac{E_{s}}{K}}{w_{k}}^{H}g_{k}$, and the noise-plus-interference term $\eta_{k}$ is a random variable with zero mean and variance $\sqrt{\frac{E_{s}}{K}}\sum_{i=1,i\neq k}^{K}|{w_{k}}^{H}g_{i}{|}^{2}+{\left\|w_{k}\right\|}^{2}$. $w_{k}$ and $g_{k}$ are the *k*-th columns of the matrices W and G, respectively.

Then, for the transmitted data of the *k*-th user, the received SINR at the BS can be expressed as
(22)$$\begin{array}{cc} \gamma_{k} & =\frac{E[|\xi_{k}x_{k}{|}^{2}]}{E[|\eta_{k}{|}^{2}]}. \end{array}$$

### 4.1. BER Upper Bound in Both Small- and Large-Scale Fading

In this letter, we consider that the transmission data of the *k*-th user is encoded using a linear convolutional code with a rate of $R_{c}=\frac{k_{c}}{n_{c}}$. Supposing that the interleaver is ideal, then the correlated bit metrics are sufficiently separated in the decoder. Through the appropriate subscript combinations, we let $x_{k}={\left[x_{k,1},\cdots ,x_{k,d}\right]}^{T}$ and ${\hat{x}}_{k}={\left[{\hat{x}}_{k,1},\cdots ,{\hat{x}}_{k,d}\right]}^{T}$, where *d* is Hamming distance of the code and ${\hat{x}}_{k,l}$ is the nearest constellation point to $x_{k,l}$. According to [25], the upper-bound for convolutionally encoded massive MIMO-MMSE systems can be evaluated using
(23)$$\begin{array}{c} P_{b}\leq \frac{1}{k_{c}}\sum_{d=d_{\mathrm{free}}}^{\infty }A_{d}f\left(d,\mu ,A\right), \end{array}$$
where $A_{d}$ is the total information weight of all error events at Hamming distance *d* and $d_{\mathrm{free}}$ is the free Hamming distance of the code. f(d,μ,A) is the Average codeword Pairwise Error Probability (APEP) that is generated by error events at Hamming distance *d*.

Similar to the analysis in [25], the pairwise error probability of two codewords can be computed as follows:(24)$$\begin{array}{cc} P\{x_{k}\to {\hat{x}}_{k}\} & =E_{\xi_{k,1},\cdots ,\xi_{k,d}}\left[Q\left(\sqrt{\frac{\sum_{l=1}^{d}{\left|\xi_{k,l}\left(x_{k,l}-{\hat{x}}_{k,l}\right)\right|}^{2}}{2E[|\eta_{k}{|}^{2}]}}\right)\right]\\ & =\frac{1}{\pi }\int_{0}^{\frac{\pi }{2}}\prod_{l=1}^{d}\underset{\Delta_{k,l}\left(\theta \right)}{\underset{⏝}{E_{\xi_{k,l}}\left[\exp\left(-\frac{|\xi_{k,l}\left(x_{k,l}-{\hat{x}}_{k,l}\right){|}^{2}}{4E[|\eta_{k}{|}^{2}]\sin^{2}\theta }\right)\right]}}d\theta , \end{array}$$
where the linear expectation is over all pairs of signal symbols $x_{k}$ and ${\hat{x}}_{k}$ whose labels differ in only one bit. Consequently, the result of the f(d,μ,A) is given by
(25)$$\begin{array}{cc} f(d,\mu ,A) & =E_{x_{k},{\hat{x}}_{k}}\left[P\{x_{k}\to {\hat{x}}_{k}\}\right]=\frac{1}{\pi }\int_{0}^{\frac{\pi }{2}}E_{x_{k},{\hat{x}}_{k}}\left[\prod_{l=1}^{d}\Delta_{k,l}\left(\theta \right)\right]d\theta \\ & \leq \frac{1}{\pi }\int_{0}^{\frac{\pi }{2}}{\left\{E_{x_{k,l},{\hat{x}}_{k,l}}\left[\Delta_{k,l}\left(\theta \right)\right]\right\}}^{d}d\theta \\ & \overset{\left(a\right)}{=}\frac{1}{\pi }\int_{0}^{\frac{\pi }{2}}{\left\{\frac{1}{\left|A\right|}\sum_{x_{k}}\sum_{{\hat{x}}_{k}}\Delta_{k}\left(\theta \right)\right\}}^{d}d\theta . \end{array}$$

In (a) of Equation (Equation 25), the subscript *l* is omitted. The $\Delta_{k}\left(\theta \right)$, as shown in Equation (Equation 24), can be written as
(26)$$\begin{array}{cc} \Delta_{k}\left(\theta \right) & =E_{\gamma_{k}}\left[\exp\left(-\frac{\gamma_{k}{\left|x_{k}-{\hat{x}}_{k}\right|}^{2}}{4E[|x_{k}{|}^{2}]\sin^{2}\theta }\right)\right]. \end{array}$$

For convenience, let the normalized distance between symbols be $c_{k}=\frac{|x_{k}-{\hat{x}}_{k}{|}^{2}}{4E[|x_{k}{|}^{2}]\sin^{2}\theta }$. Then, using (Equation 20), the form of the expectation operation in Equation (Equation 26) be changed as
(27)$$\begin{array}{cc} E_{\gamma_{k}}\left[e^{-c_{k}\gamma_{k}}\right] & =\int_{0}^{\infty }e^{-c_{k}x}f_{\gamma_{k}}\left(x\right)dx\\ & =A\sum_{m=1}^{2}\sum_{l=1}^{t_{m}}\frac{B_{m,l,t,\delta }}{(t_{m}-l)!}\int_{0}^{\infty }x^{t_{m}-l}e^{-(\delta_{m}+c_{k})x}dx\\ & =A\sum_{m=1}^{2}\sum_{l=1}^{t_{m}}\frac{B_{m,l,t,\delta }}{{\left(\frac{1}{\delta_{m}}+c_{k}\right)}^{t_{m}-l+1}}. \end{array}$$

By substituting $c_{k}$ with $\frac{|x_{k}-{\hat{x}}_{k}{|}^{2}}{4E[|x_{k}{|}^{2}]\sin^{2}\theta }$ inside Equation (Equation 27), and combining Equation (Equation 23), it holds that
(28)$$\begin{array}{c} P_{b}\leq \frac{1}{\pi k_{c}}\sum_{d=d_{\mathrm{free}}}^{\infty }A_{d}\int_{0}^{\frac{\pi }{2}}{\left[\frac{A}{\left|A\right|}\sum_{x}\sum_{\overset{˘}{x}}\sum_{m=1}^{2}\sum_{l=1}^{t_{m}}\frac{B_{m,l,t,\delta }}{{\left(\frac{1}{\delta_{m}}+\frac{|x_{k}-{\hat{x}}_{k}{|}^{2}}{4E[|x_{k}{|}^{2}]\sin^{2}\theta }\right)}^{t_{m}-l+1}}\right]}^{d}d\theta . \end{array}$$

It is obvious that Equation (Equation 25) is a function in the form of an integral. We note that a simple approximated expression can be obtained for Q(x) as [18]
(29)$$\begin{array}{c} Q\left(x\right)≃\frac{1}{12}e^{-\frac{x^{2}}{2}}+\frac{1}{6}e^{-\frac{2x^{2}}{3}}. \end{array}$$

By applying (Equation 29) into (Equation 24), Equation (Equation 24) can be simplified as
(30)$$\begin{array}{c} P\{x_{k}\to {\hat{x}}_{k}\}\\ =E_{\xi_{k,1},\cdots ,\xi_{k,d}}\left[\frac{1}{12}\exp\left(-\frac{\sum_{l=1}^{d}{\left|\xi_{k,l}\left(x_{k,l}-{\hat{x}}_{k,l}\right)\right|}^{2}}{4E[|\eta_{k}{|}^{2}]}\right)+\frac{1}{6}\exp\left(-\frac{\sum_{l=1}^{d}{\left|\xi_{k,l}\left(x_{k,l}-{\hat{x}}_{k,l}\right)\right|}^{2}}{3E[|\eta_{k}{|}^{2}]}\right)\right]\\ =\frac{1}{12}\prod_{l=1}^{d}E_{\xi_{k,l}}\exp\left(-\frac{|\xi_{k,l}\left(x_{k,l}-{\hat{x}}_{k,l}\right){|}^{2}}{4E[|\eta_{k}{|}^{2}]}\right)+\frac{1}{6}\prod_{l=1}^{d}E_{\xi_{k,l}}\exp\left(-\frac{|\xi_{k,l}\left(x_{k,l}-{\hat{x}}_{k,l}\right){|}^{2}}{3E[|\eta_{k}{|}^{2}]}\right)\\ \leq \frac{1}{12}{\left[E_{\gamma_{k}}\exp\left(-\frac{\gamma_{k}{\left|\left(x_{k}-{\hat{x}}_{k}\right)\right|}^{2}}{4E[|x_{k}{|}^{2}]}\right)\right]}^{d}+\frac{1}{6}{\left[E_{\gamma_{k}}\exp\left(-\frac{\gamma_{k}{\left|\left(x_{k}-{\hat{x}}_{k}\right)\right|}^{2}}{3E[|x_{k}{|}^{2}]}\right)\right]}^{d}. \end{array}$$

For the convenience of calculation, we define $\frac{|\left(x_{k}-{\hat{x}}_{k}\right){|}^{2}}{4E[|x_{k}{|}^{2}]}$ as $b_{1}$, and $\frac{|\left(x_{k}-{\hat{x}}_{k}\right){|}^{2}}{3E[|x_{k}{|}^{2}]}$ as $b_{2}$. Similar to that in Equation (Equation 25), we have
(31)$$\begin{array}{cc} f(d,\mu ,A) & =E_{x_{k},{\hat{x}}_{k}}\left[P\{x_{k}\to {\hat{x}}_{k}\}\right]\\ & =\frac{1}{12}{\left[\frac{1}{\left|A\right|}\sum_{x_{k}}\sum_{{\hat{x}}_{k}}E_{\gamma_{k}}\left[e^{-\gamma b_{1}}\right]\right]}^{d}+\frac{1}{6}{\left[\frac{1}{\left|A\right|}\sum_{x_{k}}\sum_{{\hat{x}}_{k}}E_{\gamma_{k}}\left[e^{-\gamma b_{2}}\right]\right]}^{d}\\ & =\frac{1}{12}{\left[\frac{A}{\left|A\right|}\sum_{x_{k}}\sum_{{\hat{x}}_{k}}\sum_{m=1}^{2}\sum_{l=1}^{t_{m}}\frac{B_{m,l,t,\delta }}{{\left(\frac{1}{\delta_{m}}+b_{1}\right)}^{t_{m}-l+1}}\right]}^{d}\\ & +\frac{1}{6}{\left[\frac{A}{\left|A\right|}\sum_{x_{k}}\sum_{{\hat{x}}_{k}}\sum_{m=1}^{2}\sum_{l=1}^{t_{m}}\frac{B_{m,l,t,\delta }}{{\left(\frac{1}{\delta_{m}}+b_{2}\right)}^{t_{m}-l+1}}\right]}^{d}. \end{array}$$

Now, by replacing $b_{1}$ and $b_{2}$ with $\frac{|\left(x_{k}-{\hat{x}}_{k}\right){|}^{2}}{4E[|x_{k}{|}^{2}]}$ and $\frac{|\left(x_{k}-{\hat{x}}_{k}\right){|}^{2}}{3E[|x_{k}{|}^{2}]}$ respectively, and combining Equations (Equation 31) and (Equation 23), the BER upper bound for the coded systems is given by
(32)$$\begin{array}{cc} P_{b}\leq \frac{1}{\pi k_{c}}\sum_{d=d_{\mathrm{free}}}^{\infty }A_{d} & \{\frac{1}{12}{\left[\frac{A}{\left|A\right|}\sum_{x_{k}}\sum_{{\hat{x}}_{k}}\sum_{m=1}^{2}\sum_{l=1}^{t_{m}}\frac{B_{m,l,t,\delta }}{{\left(\frac{1}{\delta_{m}}+\frac{|\left(x_{k}-{\hat{x}}_{k}\right){|}^{2}}{4E[|x_{k}{|}^{2}]}\right)}^{t_{m}-l+1}}\right]}^{d}\\ & +\frac{1}{6}{\left[\frac{A}{\left|A\right|}\sum_{x_{k}}\sum_{{\hat{x}}_{k}}\sum_{m=1}^{2}\sum_{l=1}^{t_{m}}\frac{B_{m,l,t,\delta }}{{\left(\frac{1}{\delta_{m}}+\frac{|\left(x_{k}-{\hat{x}}_{k}\right){|}^{2}}{3E[|x_{k}{|}^{2}]}\right)}^{t_{m}-l+1}}\right]}^{d}\}. \end{array}$$

### 4.2. Asymptotical Diversity Order

Now, we are interested in further analyzing the derived BER expression to find the diversity of the coded large-scale multiuser MIMO systems with MMSE receivers. In the high-SNR region, the asymptotical diversity order is defined as:(33)$$\begin{array}{cc} D & =-\lim_{\mathrm{SNR}\to \infty }\frac{\log_{10}P_{b}}{\log_{10}\mathrm{SNR}}\\ & ≃-\lim_{\mathrm{SNR}\to \infty }\frac{\log_{10}f(d_{free},\mu ,A)}{\log_{10}\mathrm{SNR}}. \end{array}$$

Next, we analyze the various parameters that make up $f(d_{free},\mu ,A)$. Since $\mathrm{SNR}=\frac{E_{s}\mathrm{tr}\left(P\right)}{KN_{0}}$, $\rho =\frac{KN_{0}}{E_{s}}$, $\delta_{1}=\zeta_{k}G$ and $\delta_{2}=\zeta_{k}/\rho$, both 1/ρ and $\delta_{2}$ are proportional to SNR. When SNR approaches infinity, $\delta_{1}$ approaches a constant. Therefore, in the high SNR region, the parameter *A* is proportional to ${\mathrm{SNR}}^{-(R-K+1)}$, and the parameter $B_{m,l,t,\delta }$ can be regarded as a constant. In addition, when the distance between the constellation points is fixed, the SNR tends to infinity, which is equivalent to the noise power tending to zero. Then, the denominator part of Equation (Equation 27) can be treated as a constant. The essence of the integration is a summation, without changing the exponential portion of the parameters associated with SNR. From the above analysis, we can see that, in the high SNR region, $f(d_{free},\mu ,A)$ is proportional to ${\mathrm{SNR}}^{-d_{free}\left(R-K+1\right)}$, and then combined with Equation (Equation 33), the asymptotical diversity order $D=d_{free}\left(R-K+1\right)$ is obtained.

## 5. Numerical Results and Simulations

In this section, numerical results and simulations are performed to validate the accuracy of the theoretical analysis. We consider a practical scenario where a BS equipped with *R* receiving antennas is located at the center of a hexagonal cell with a radius of 1 km. We assume that K=5,10 users are distributed independently and uniformly at random in the cell and the distance between each user and the base station is greater than $r_{h}=100$ meters, that is, for the *k*-th user, $r_{k}>r_{h}$. The largescale fading parameter $\zeta_{k}$ associated with geometric attenuation and shadow fading can be estimated by $\zeta_{k}=\tau_{k}/{\left(r_{k}/r_{h}\right)}^{\upsilon }$, where $\tau_{k}>0$ denotes the shadow fading and obeys $10\log_{10}\left(\tau_{k}\right)∼N\left(0,\sigma_{sf}^{2}\right)$. υ represents the pathloss exponent. In the simulation, we choose $\sigma_{sf}^{2}=8$ dB and υ=3.8. Furthermore, the gray labeled 16-QAM constellation is considered and the statistical model of the channel is assumed to be perfectly known at the receiver.

In each of the pictures, the left-hand side is the case of the conventional multiuser MIMO, and the right-hand side corresponds to the large-scale multiuser MIMO scenes. We can see that, in all cases, the theoretical curves of our approximate approaches are well fit to the simulations.

We first provide results for the PDF of SINR based on linear MMSE detection for multiuser MIMO systems in Figure 1, where SNR = 10 dB. Obviously, for a fixed number of users *K*, when the number of receiving antennas of the BS increases, the mean value of the SINR also increases, and it is more capable of providing a larger value of SINR. Therefore, the performance of the system tends to be better. Figure 2 presents the SEP performance of 16 QAM, and 64 QAM signaling for an uncoded Large-Scale MIMO system with an MMSE receiver. It should be explained that, based on the derived SINR distribution, the corresponding SEP expression can be obtained by using the method of [16]. The black curves correspond to the system used in [16], where only Rayleigh fading channels were considered. By comparison, it can be seen that large-scale fading has a greater impact on the performance of the system. However, which must be considered in a practical multi-user scenario. In Figure 3, we plot the upper-bound on BER of of 16-QAM signaling scheme, for the encoded MMSE–MIMO systems, where the half rate ${\left(133,171\right)}_{8}$ convolutional code was used and the free distance is $d_{f}=10$. We can see that, in the high-SNR region, our theoretical curves perfectly tend to the experimental curves. In the low SNR region, the anti-interference ability of the system under consideration is relatively weak, and the desired signal is severely affected by the interference. Therefore, the value of PEP is relatively large, and the BER upper-bound obtained based on PEP is relatively loose. It is not difficult to observe that, when the number of users *K* and the number of receiving antennas *R* of the BS have been given, the BER performance deteriorates as the modulation order increases. From another perspective, QAM,16 QAM and 64 QAM require a higher SNR than binary phase shift keying (BPSK) for a given BER. Furthermore, the system performance is deteriorated as the difference between the number of transmit and receive antennas decreases. This is because, when SNR grows larger, the encoded MMSE–MIMO receiver realizes an asymptotic diversity of $d_{free}\left(R-K+1\right)$. Since asymptotic diversity (Equation 33) requires SNR to be infinite, the simulations in the actual SNR region deviate from the theoretical result (Equation 33). In addition, since the ZF algorithm ignores the effects of noise, the MMSE exhibits better BER performance than the ZF algorithm over the entire SNR region. However, as the number of antennas continues to increase, the performance of ZF and MMSE algorithms is similar.

## 6. Conclusions

This work focuses on the uplink performance of massive multiuser MIMO networks. We proposed a novel approach to derive the distribution of the SINR for an MMSE decoder. Then, we derived the PEP, which was used to obtain the result of the upper-bound on BER for convolutionally encoded massive MMSE–MIMO systems. Finally, the results of Monte-Carlo simulations show that our analytical expressions are valid.

## Figures and Tables

**Figure 1 sensors-19-02884-f001:**
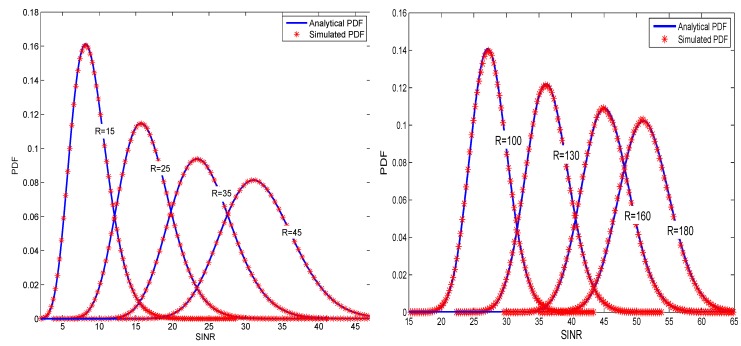
PDF of the SINR based on linear MMSE detection for multiuser MIMO systems at SNR = 10 dB. The choice of the number of users is 5 and 10, and the number of BS receiving antennas is from 15 to 180.

**Figure 2 sensors-19-02884-f002:**
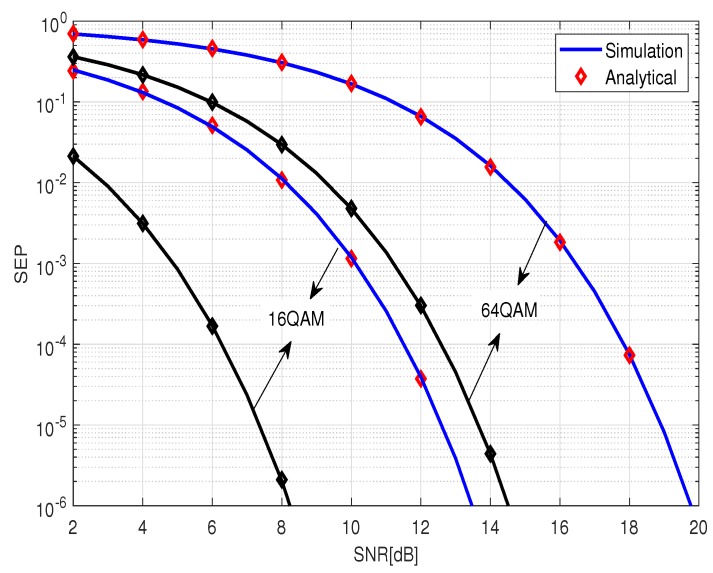
Symbol error probability (SEP) performance of 16 QAM, and 64 QAM signaling for an uncoded large-scale MIMO system equipped with a linear MMSE equalizer. The number of users *K* and the number of receiving antennas *R* at the base station are 10 and 200, respectively.

**Figure 3 sensors-19-02884-f003:**
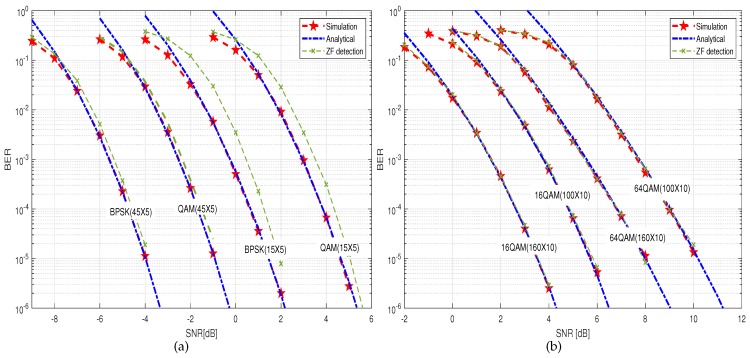
Upper-bound on BER of various signaling schemes for convolutionally encoded multiuser MMSE-detected MIMO system (**a**) the number of users *K* is set to 5, and the number of BS receiving antennas R is set to 15, 45; (**b**) the number of users is set to 10, and the number of BS receiving antennas *R* is set to 100, 160.

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
