# Peer review of "Accurate Performance Analysis of Coded Large-Scale Multiuser MIMO Systems with MMSE Receivers"

_sensors, 2019, doi:10.3390/s19132884_

Round 1
Reviewer 1 Report
This paper derives SINR expression for massive MU-MIMO with MMSE receiver.
The contribution of this paper in sec. 1 should be rewritten. The focus should be comparison to the state of the art such as (especially) the authors’ another paper [14] (conventional MIMO=>massive MIMO, uncoded=>coded?) and others such as
"Uplink Performance Analysis of Mixed-ADC Massive MIMO Systems with MMSE Receivers." 2018 10th International Conference on Wireless Communications and Signal Processing (WCSP). IEEE, 2018.
Making Cell-Free Massive MIMO Competitive With MMSE Processing and Centralized
Implementation." arXiv preprint arXiv:1903.10611 (2019).
Coded performance analysis in (23)-(26) looks like standard convolutional coded bit error probability upper bound, which could be found in error correction code textbooks. What is the significant contribution of it?
In Sec. 1, 1st paragraph, more recent references about MU-MIMO and massive MU-MIMO should be added such as
"A Near Optimal Scheduling Algorithm for Efficient Radio Resource Management in Multi-user MIMO Systems." Wireless Personal Communications (2019): 1-17.
"Average PSNR optimized cross layer user grouping and resource allocation for uplink MU-MIMO
OFDMA video communications." IEEE Access6 (2018): 50559-50571.
"Fast iterative WSVT algorithm in WNN minimization problem for multiuser massive MIMO channel
estimation." International Journal of Communication Systems 31.1 (2018): e3378
"Sum Ergodic Capacity Analysis Using Asymptotic Design of Massive MU-MIMO Systems." Wireless Personal Communications 100.4 (2018): 1743-1752.
Author Response
Thank you very much for your suggestions.
In the uploaded PDF, we have provided a point-by-point response to all comments.

Reviewer 2 Report
This study presents the derivation of the analytical expression for the analysis of the uplink performance of the massive multiuser MIMO network.
1) What would be the effect of different modulation schemes (BPSK, QPSK, 16-QAM, and 64-QAM, etc.) on the performance of the analytical expression of the BER/SINR? Present the comparison results.
2) What would be the diversity order with the proposed approach? What would be the effect of the derived expression for the upper-bound on the diversity order of the coded MIMO system with MMSE receiver? Present the comparison of the diversity order analytical and simulation results.
3) Add more details about the simulation setting.
4) What is the advantage of the used MMSE approach over ZF and Effective Matrix Inversion for the massive MIMO system? Present the comparison.
5) Add a comparison of the proposed approach with the state of the art approaches.
6) Explain the reason for the mismatch between the theoretical curves and experimental curves for the low-SNR region.
Author Response
Thank you very much for your comments.
In the uploaded PDF, we have provided a point-by-point response to all comments.

Round 2
Reviewer 1 Report
No further comments
Reviewer 2 Report
Paper has been revised as per suggestions.